# Interaction of Diet and Ozone Exposure on Oxidative Stress Parameters within Specific Brain Regions of Male Brown Norway Rats

**DOI:** 10.3390/ijms20010011

**Published:** 2018-12-20

**Authors:** Joseph M. Valdez, Andrew F. M. Johnstone, Judy E. Richards, Judith E. Schmid, Joyce E. Royland, Prasada Rao S. Kodavanti

**Affiliations:** 1Neurotoxicology Branch, Toxicity Assessment Division, NHEERL/ORD, U.S. Environmental Protection Agency, Research Triangle Park, NC 27711, USA; jvald010@ucr.edu (J.M.V.); johnstone.andrew@epa.gov (A.F.M.J.); Schmid.judy@epa.gov (J.E.S.); 2Environmental Public Health Division, NHEERL/ORD, U.S. Environmental Protection Agency, Research Triangle Park, NC 27711, USA; Richards.Judy@epa.gov; 3Integrated Systems Toxicology Division, NHEERL/ORD, U.S. Environmental Protection Agency, Research Triangle Park, NC 27711, USA; royalj@yahoo.com

**Keywords:** high-fructose, high-fat, ozone, oxidative stress, neurotoxicity, total antioxidants, reactive oxygen species, protein carbonyls

## Abstract

Oxidative stress (OS) contributes to the neurological and cardio/pulmonary effects caused by adverse metabolic states and air pollutants such as ozone (O_3_). This study explores the interactive effects of O_3_ and diet (high-fructose (FRUC) or high–fat (FAT)) on OS in different rat brain regions. In acute exposure, there was a decrease in markers of reactive oxygen species (ROS) production in some brain regions by diet and not by O_3_. Total antioxidant substances (TAS) were increased in the cerebellum (CER) and frontal cortex (FC) and decreased in the striatum (STR) by both diets irrespective of O_3_ exposure. Protein carbonyls (PC) and total aconitase decreased in some brain regions irrespective of exposure. Following subacute exposure, an increase in markers of ROS was observed in both diet groups. TAS was increased in the FC (FAT only) and there was a clear O_3_ effect where TAS was increased in the FC and STR. Diet increased PC formation within the CER in the FAT group, while the hippocampus showed a decrease in PC after O_3_ exposure in controls. In general, these results indicate that diet/O_3_ did not have a global effect on brain OS parameters, but showed some brain region- and OS parameter-specific effects by diets.

## 1. Introduction

Over the past few decades, there has been an increase in both diabetes and obesity throughout the world, with steeper trends seen in more developed countries that have adopted the Western diet rich in refined sugars and saturated fats (SF) [1,2]. Research into the effects of consuming these diets has revealed a strong correlation with severe metabolic disorders such as type 2 (non-insulin-dependent) diabetes and obesity [3]. The rise in obesity strongly correlates to the increase in the use of high-fructose corn syrup in sweetened beverages and foods, with the U.S. being one of the largest consumers [4]. It has also been argued that the increased consumption of SF has contributed significantly to the rise in obesity [5]. Though there is controversy over whether dietary fat, sugar, or even economic status (causing a reliance on high caloric food and beverages) is the primary contributor to adverse metabolic states, there is agreement that energy-dense foods (high in fat and carbohydrate content) lead to severe metabolic disorders [5,6,7]. 

A common adverse outcome of diabetes and obesity is damage to cardiac, renal, nervous, and/or hepatic tissues. A potential mechanism is hyperglycemic-induced production of reactive oxygen species (ROS) which can, in turn, cause oxidative stress (OS)-induced tissue damage [8]. For instance, a diet rich in fat has been shown to lead to OS within the hepatic tissue of rats [9]. Recent studies have shown that inhalation of airborne toxicants can exacerbate the effects of metabolic disorders. Bass et al. [10] have shown that rats exposed to 1.0 ppm ozone (O_3_) consistently develop glucose intolerance across all life stages. Interestingly, O_3_ has also been shown to cause activation of the hypothalamic–pituitary–adrenal (HPA) axis, a keystone pathway in sympathetic nervous system (SNS) activation [11]. It is well established that the SNS can exacerbate and possibly causes physiological symptoms associated with obesity, metabolic syndrome, and diabetes by causing insulin resistance, increased heart rate and blood pressure, and increased lipolysis in visceral fat [12,13].

O_3_ is a prototypic air pollutant and a major component of smog. It is one of the six criteria air pollutants and the known health effects include lung injury and nervous system effects [14,15]. Recent studies indicate that O_3_ exposure leads to pulmonary immune response resulting in circulating factors in blood (microglial proinflammatory response and β-amyloid 42 neurotoxicity) and these signals are detected by microglia in the brain resulting in primed proinflammatory phenotypes, suggesting that the lung-brain axis has significant implications for how air pollution may affect the brain to augment CNS effects [16]. In addition, O_3_ has been reported to elevate cytokines and causes OS in the brain [17]. Most studies have focused on the adverse effects of diet or various stressors alone and few reports exist about combined effects of diet and environmental stressors. In either case, the organism’s physiological endpoints (i.e., behavioral, pulmonary, cardiac, metabolism, etc.) are the main focus [18,19]. There are few, if any, studies related to the possible interaction between diet and O_3_, especially on brain neurochemistry. Studies using the same cohort of rats showed that treatment with high-fructose and high-fat diets attenuated some O_3_-induced effects on pulmonary function, neurobehavior, and metabolism [20]. The purpose of this study is to further understand this interaction by exploring the potential for high-fat or high-fructose effects on OS within specific brain regions of animals exposed to O_3_ either acutely or subacutely. Of the selected brain regions, the cerebellum (CER) is involved in motor activity while the hypothalamus (HYP) is involved in the pulmonary and cardiac effects through the HPA axis. Combining the diet and O_3_ treatments will provide insight into a more realistic exposure paradigm of individuals who live in areas with high amounts of O_3_ pollution and who have access to highly caloric foods.

## 2. Results

### 2.1. Production of Reactive Oxygen Species (ROS)

*Acute Exposure:* In order to measure the production of ROS, NQO1 and UBIQ-RD activities were assessed. The striatum (STR) showed a significant diet effect (F_2,24_ = 4.14, p = 0.028) for the NQO1 measure, as there was a decrease in activity with both diets when compared to the regular (5001) diet (*p* < 0.05; Figure 1A). The UBIQ-RD activity showed a significant interaction between diet and O_3_ treatment (F_2,24_ = 9.63, *p* = 0.001) in the hippocampus (HIP). The FRUC/O_3_- exposed group and the FAT-unexposed group both had higher UBIQ-RD than the FRUC-unexposed group (*p* < 0.05; Figure 2A).

*Subacute Exposure:* The NQO1 measure within the subacute group did reveal a diet effect in the CER (F_2,26_ = 9.16, *p* = 0.001), with all three diets being significantly different from each other, in the order: FRUC > regular > FAT (Figure 1B). In the FC, while the O_3_–diet interaction was not quite significant (F_2,24_ = 3.21, *p* = 0.058), the O_3_-exposed regular diet group had higher NQO1 activity than the exposed FAT group (*p* < 0.05).

The UBIQ-RD data showed a diet effect in both the FC (F_2,26_ = 7.24, *p* = 0.003) and the HIP (F_2,26_ = 3.79, *p* = 0.036). In the FC, the FRUC group had higher UBIQ-RD than both FAT and regular diet groups (*p* < 0.01); in the HIP, FAT and FRUC were higher than the regular diet (Figure 2B, *p* < 0.05).

### 2.2. Antioxidant Homeostasis

*Acute Exposure:* Overall, there were only a few occasions where O_3_ exposure led to a significant change in antioxidant homeostasis, while dietary condition led to multiple instances of significance. In the FC, across the dietary groups, O_3_ (0.8 ppm) exposure was associated with a significant increase in TAS (F_1,26_ = 7.66, *p* = 0.010). With respect to dietary treatment, we found that the CER (F_2,26_ = 5.04, *p* = 0.014) showed a significant increase in total antioxidants in both the FRUC (*p* < 0.01) and the FAT (*p* < 0.01) diets relative to the regular diet. In contrast, TAS in the STR (diet F_2,26_ = 6.039, *p* = 0.007) showed a significant decrease in both the FRUC (*p* < 0.05) and the FAT (*p* < 0.01) diets when compared to the Regular (5001) diet (Figure 3A).

A significant interaction was found between diet and O_3_ treatment within the FC for the γ-GCS measure (F_2,24_ = 4.672, *p* = 0.019), but this was not supported by any significant differences among the diet and/or O_3_ groups. We did observe a decrease in γ-GCS in the HIP (F_2,26_ = 13.77, *p* < 0.001) for both FRUC and FAT groups relative to the regular diet group (*p* < 0.01), while in the STR (F_2,26_ = 9.187, *p* = 0.001) there was a significant decrease in the FAT group relative to both the high-fructose and the regular diets (*p* < 0.01; Figure 4A) in both air- and O_3_-exposed rats.

*Subacute Exposure:* The subacute group did show a significant interaction between dietary condition and O_3_ treatment in the FC (F_2,26_ = 6.275, *p* = 0.006) for TAS where there was also a significant diet effect (F_2,24_ = 6.00, *p* = 0.006). Across O_3_ groups, the FAT diet was associated with higher TAS than the regular diet (*p* = 0.008); however, among the individual diet by O_3_ groups, only FAT/control was significantly higher than the two Regular Diet/O_3_ groups (*p* < 0.05). FAT/control was also significantly higher in TAS than the FRUC/control group in FC.

In the STR, there was a significant O_3_ treatment effect in TAS, with higher TAS levels in the O_3_-exposed animals (F_2,24_ = 4.84, *p* = 0.037). Interestingly, there was a significant O_3_ treatment effect in the γ-GCS measure within the CER (F_1,26_ = 6.28, *p* = 0.019) as well, where O_3_ was associated with lower γ-GCS activity (Figure 3B and Figure 4B).

### 2.3. Oxidative Damage

*Acute Exposure:* Oxidative damage was determined by measuring PC and aconitase activity. Within the acute group, the only brain region that showed any significant changes in PC content resulting from O_3_ treatment was the CER, which had both an O_3_ effect (F_1,26_ = 5.73, *p* = 0.024) and a diet treatment effect (F_2,26_ = 4.223, *p* = 0.026). Across diet groups, there was a decrease following exposure to O_3_ (*p* < 0.05). With respect to differences among diet groups, PC content showed a significant decrease in the FRUC group relative to both FAT and regular diet (*p* < 0.05; Figure 5A).

The measure of aconitase activity revealed a significant O_3_ by diet interaction within the CER (F_2,24_ = 8.89, *p* = 0.001). This was not due to a systematic O_3_ or diet effect; however, the FRUC/control and Regular diet/O_3_ groups had significantly higher aconitase than the Regular diet/control group (*p* < 0.05). Within the HIP, there were differences in aconitase activity due to diet (F_2,26_ = 7.023, *p* = 0.004); there was a significant decrease in the FAT and FRUC animals relative to the regular diet groups (*p* ≤ 0.01). Within the STR (diet F_2,26_ = 13.44, *p* < 0.001), there was also a significant decrease in aconitase activity in both the FRUC and FAT diets relative to the regular diet (*p* < 0.01; Figure 6A).

*Subacute Exposure:* In the subacute group, there was a significant interaction between diet and O_3_ in PC in the HIP (F_2,22_ = 6.18, *p* = 0.007). This was due to an overall O_3_ effect (F_1,22_ = 10.18, *p* = 0.004) where across the three diet groups, O_3_ treatment was associated with a decrease in PC levels; however, individually, the O_3_-treated FRUC group had a small (nonsignificant) increase in PC, while regular and FAT diets showed a decrease. Of the latter two, only the regular diet decrease was individually significant (Figure 5B; *p* < 0.05) in HIP. There was a significant diet effect in the CER (F_2,32_ = 9.11, *p* = 0.001) where FAT animals had significantly higher PC levels than animals fed FRUC (*p* < 0.05) and regular diet (*p* < 0.01).

There was a significant decrease in aconitase activity across diet groups when compared to their respective air controls for both HIP and HYP (F_1,26_ = 7.80, 7.13, with *p* = 0.010, 0.013, respectively). In the STR, there was a significant diet effect (F_2,26_ = 4.68, *p* = 0.018) where the FAT animals had lower aconitase levels than those on the regular diet (Figure 6B). There was also a diet effect in the CER (F_2,26_ = 4.68, *p* = 0.018) where all three diet groups were different from each other, with the order of aconitase activity being regular > high-fructose > high-fat (*p* < 0.001 for regular vs. FAT, and *p* < 0.05 for the other two comparisons).

## 3. Discussion

In this study, we attempt to understand the combined effects of O_3_ exposure and different diets on oxidative stress (OS) parameters in different brain regions of rats. The diets (FRUC and FAT) chosen are thought to be associated with adverse metabolic states and are also of concern in compounding effects to environmental toxicant exposure [20]. In short, juvenile rats were given either a control, high-fat, or high-fructose diet and exposed to either air (control) or 0.8 ppm O_3_ in acute or subacute exposure paradigms. Multiple brain regions (cerebellum, frontal cortex, hippocampus, hypothalamus, and striatum) were examined for markers of OS, specifically, oxyradical production (NQO1 and UBIQ-RD), antioxidant homeostasis (TAS and γ-GCS), and oxidative damage (protein carbonyls and aconitase).

NQO1 (NAD(P)H: quinone oxidoreductase) is an electron reductase enzyme found specifically in the cytosol and is expressed in many tissues [21]. In brain tissue, NQO1 is found abundantly in astrocytes and endothelial cells, is neuroprotective against oxidative damage in vivo and in vitro, and has increased activity in neurodegenerative diseases, such as Parkinson’s and Alzheimer’s [22,23,24]. Ubiquinone reductase (UBIQ-RD) is the first complex of the electron transport chain found in the inner membrane of the mitochondria and is crucial for ATP production [25] and also plays a role in neurodegenerative diseases [26,27,28,29]. There was not an apparent global O_3_/diet interaction when compared to controls for markers of ROS production. Only the HIP showed a diet/O_3_ interaction in UBIQ-RD activity after acute O_3_ exposure. However, there was a diet-driven effect in the acute FRUC and FAT groups in the STR region of subjects when compared to controls indicated by a decrease in NQO1 activity. Conversely, there was an increase in activity in the subacute CER subjects with both test diets. Age may play a factor in these results, as the subacute group was four weeks older than the acute group when sacrificed and previous work has shown that age may be a factor in OS in differing brain regions of Brown Norway rats [30]. Effects on UBIQ-RD were, again, largely due to diet. We found the FRUC versus control diet (5001) to increase activity in the HIP following acute O_3_ exposure but decreased activity in the air controls. The FRUC diet also produced changes in UBIQ-RD activity with the longer subacute exposure regimen, showing significant increases in the FC and HIP in both the air- and O_3_-exposed animals. The FAT diet increased UBIQ-RD activity in the HIP under these conditions as well. The electron transport chain is known to be a source of oxygen radical production [30]. Changes in UBIQ-RD activity due to changes in resource input could impact energy metabolism with a downstream consequence on ROS production.

Protective antioxidant measures were also affected in some specific brain regions. Antioxidants are important mitigators of ROS damage in all tissues, including the brain, and their activity is highly influenced by the cause and region of damage [31]. TAS and γ-GCS were measured in the same brain regions. TAS measures allow insight as to the amount of total antioxidants available within a cell or tissue and indicate the capacity of the cell to combat ROS. Acutely exposed rats showed a diet-driven effect in both CER (increase) and STR (decrease), while in FC, O_3_ caused an increase in TAS regardless of diet. Subacutely, diet seemed to be the driving factor for changes in TAS, with an increase in FC of FAT air and both experimental diets in O_3_ subjects. An O_3_ effect was seen in all diets within the STR. γ-GCS enzyme activity indirectly measures the abundance of glutathione, as it is the rate-limiting step in its production. Glutathione, being a potent antioxidant, was able to prevent damage from free radical productions and the amount measured gives insight as to the amount of ROS present. Again, diet and not O_3_ was the main factor that altered γ-GCS activity and was only seen in the acute-exposure animal HIP (FAT and FRUC) and STR (FAT only). It should be mentioned that following acute exposure, when all groups were factored in, there was an interaction between dietary and O_3_ treatments in the FC, and the same was also found in the subacute CER groups. Without a clear trend in antioxidant substance as compared to ROS production, it is difficult to conclude that diet has any effect on OS that may be O_3_-derived.

When oxyradical production is unable to be controlled, oxidative damage ensues. Total aconitase and protein carbonyls were assessed and measured as metrics of oxidative damage following O_3_ exposure. Aconitase is an enzyme that is both cytosolic and mitochondrial, with the major amount being in the mitochondria, and is vulnerable to free radical damage. Cytosolically, aconitase plays a critical role in iron metabolism, whereas in the mitochondria, it is a key player in mtDNA stabilization, which makes it an indirect indicator of mtDNA damage, but both have been linked to neurodegenerative diseases [32,33]. In this study, diet was again the driving factor both with acute and subacute O_3_ exposure, with a decrease in aconitase activity in affected brain regions (acutely—HIP and STR, subacutely—CER and STR). There was no direct pattern seen with protein carbonyls—a strong indicator of oxidative damage through oxidized proteins via Lysine, Arginine, Proline, and Threonine residues [34]. Acutely, CER showed a decrease in all groups exposed to O_3_, but diet remained the main factor in the FRUC group. Following longer exposure times, in the CER, diet showed a significant increase in the FAT group while the HIP revealed a possible O_3_/diet interaction within the FRUC groups. The only other subacute measure that would be predictive of oxidative damage was an increase in UBIQ-RD activity. Interestingly, acutely, there was a decrease in γ-GCS activity in the HIP. This may give insight that initial O_3_ exposure starts an oxidative damage pathway in the HIP region as shown by Rivas-Arancibia et al. [35] during chronic exposure.

Before concluding, the diets themselves must be addressed. In an effort to understand the outcomes of this study, we looked at the diets themselves as they were chosen based on equal caloric content, not equal nutritional makeup (i.e., minerals and vitamins). Both the high-fructose and high-fat diets were purified, meaning they lacked the natural phytoestrogens and isoflavones found in the control diet. This possesses the possibility that the lack of natural antioxidants potentially contributed to the results seen in this study. Other rodent (rat/mouse) studies with matched diets [36,37,38,39] have shown similar OS results as have studies not using matched diets [40,41] and those where diet composition was not reported [42,43]. In this three-tiered attempt to identify an increased susceptibility to metabolic disorders due to an interaction of calorie-dense diets and O_3_ exposure, cause and effect remain elusive. Parameters measured in this current study inferred OS from free radical production, antioxidant potential, and macromolecule damage in animals with differing diets in acute and subacute paradigms. In a separate study from this same cohort of animals, treatment with either diet led to some improvement of O_3_ effects on ventilatory, behavioral, and metabolic endpoints. In male Brown Norway rats, tidal volume and breathing frequency (high-fat only, subacute) were somewhat improved, while Penh (a metric of ventilatory resistance) and minute ventilation were exacerbated (both high-fat and high-fructose, subacute only). Vertical and horizontal exploratory (subacutely) behaviors where improved in both high-fat and high-fructose diets, while high-fructose diet alone gave some metabolic improvement in cholesterol and triglyceride serum levels [20]. O_3_ treatment alone led to significant increases in pulmonary pathology [20] with an increase in bronchoalveolar lavage fluid eosinophil counts (high-fat and high-fructose, subacute). Overall, OS parameters measured in this and related studies [20,44] do not suggest a consistent interaction between O_3_ and diet on oxidative damage pathways (for schematic of global subacute effects, see Figure 7). They do provide insight as to how highly caloric diets could affect neuronal OS, which will be useful in correlating lifestyle choices in relation to neurodegenerative diseases.

## 4. Materials and Methods

### 4.1. Animals

The use of Brown Norway (BN) rats was described previously by Gordon et al. [45,46]. In short, male BN rats were acquired at postnatal day (PND) 22 and maintained in an AAALAC-approved, pathogen-free (SPF; Charles River Laboratories, Raleigh, NC, USA) facility. They were housed individually in polycarbonate cages containing heat-treated beta chip bedding (Nepco, Warrensburg, NY, USA) with enviro-dry added for nest building (Shepherd Specialty Papers, Creedmoor, NC, USA) and held at a constant temperature of 22 °C, on a 12 h light and 12 h dark cycle. All procedures were approved by the NHEERL Institutional Animal Care and Use Committee (LAPR # 18-06-002 with approval date on 12 August 2016), which ensures conformance with the 1996 NRC “Guide for the Care and Use of Laboratory Animals”.

### 4.2. Dietary Regimen

Access to food (Rodent Chow 5001: Ralston Purina Laboratories, St. Louis, MO, USA) and tap water were available ad libitum unless otherwise stated. Protocols were approved by the NHEERL Institutional Animal Care and Use Committee prior to initiation of this study, and adhered to National Academy of Sciences guidelines on the use of laboratory animals in research. At PND 27, the rats were allocated to one of three dietary (balanced by caloric content) groups for the remainder of the study: regular (Purina 5001), high-fructose (FRUC), or high-fat (FAT) diets. The FRUC and FAT diets were commercially prepared (Harlan, Madison, WI, USA), with 60% of calories from FRUC (containing 600 g/kg fructose; 50 g/kg lard: TD.89247) or FAT (containing 310 g/kg of lard; 90 g/kg sucrose: TD.06414; Table 1).

### 4.3. O_3_ Generation and Exposure

O_3_ generation and exposure paradigm were described previously [20]. Briefly, O_3_ was generated from oxygen by a silent arc discharge generator (OREC, Phenix, AZ, USA), and its entry into the Rochester-style Hinner chambers was controlled by mass flow controllers. Monitoring of O_3_ concentration within the chambers was accomplished via photometric O_3_ analyzers (API Model 400, Teledyne, San Diego, CA, USA). During exposure, the air temperature and relative humidity inside the four chambers (two control and two O_3_ chambers) were monitored hourly. All air entering the control and O_3_ chambers was filtered with HEPA 0.3-micron filters rated at 99.97% efficient. Stainless steel wire exposure cages were used to house individual rats. Each individual cage was part of a 16-cage unit.

Following 16 weeks on their respective dietary regimen (control, high-fructose, or high-fat, *n* = 10 for each diet), rats were exposed to O_3_ either acutely or subacutely. In the acute study, rats were exposed once to filtered air or 0.8 ppm O_3_ for 5 h and then sacrificed the next day (18 h postexposure). In the subacute study, rats were exposed to filtered air or O_3_ (0.8 ppm) for 5 h/d, 1 d/week for four consecutive weeks and sacrificed 18 h postexposure (Figure 8).

### 4.4. Necropsy and Tissue Isolation

Rats were anesthetized with an overdose of sodium pentobarbital (Virbac AH, Inc., Fort Worth, TX, USA; 50–100 mg/kg, ip). Brains were then quickly removed and brain regions (frontal cortex (FC), cerebellum (CER), striatum (STR), hippocampus (HIP), and hypothalamus (HYP)) were dissected on ice [47], quick-frozen on dry ice, and stored at −80 °C until analyzed.

### 4.5. Tissue Preparation

The brain tissues were weighed, homogenized with a polytron in 20 mM Tris-HCl buffer (pH 7.4) at 50 mg/mL, and centrifuged at 8000× *g* for 20 min. The supernatants were assayed for the selected OS markers including ROS production, antioxidant homeostasis, and oxidative damage. All assays, except protein carbonyl content, were modified and adapted for use on the KONELAB clinical chemistry analyzer (Thermo Clinical LabSystems, Espoo, Finland). For measures of protein carbonyl content, brain tissues were weighed, homogenized in 50 mM phosphate buffer containing 1mM EDTA (pH 6.7) at 50 mg/mL, and centrifuged at 10,000× *g* for 15 min at 4 °C. Protein contents were determined using either a Coomassie Plus Protein Assay Kit or a Pierce BCA Protein Assay Kit (Pierce, Rockford, IL, USA).

### 4.6. Markers of ROS Production

NAD(P)H:quinone oxidoreductase (NQO1) and NADH-ubiquinone reductase (UBIQ-RD) were selected as markers of ROS production as they play critical roles in several neurodegenerative diseases in which OS is a potential responsible pathway. NQO1 activity was assayed by measuring the NADH and menadione-dependent, dicumarol-inhibited reduction of cytochrome C [48,49]. NQO1 activity was calculated from the difference in reaction rates obtained with and without dicumarol. An extinction coefficient of 18.5 mM-1cm-1 was used in calculations of specific activity [49].

UBIQ-RD activity was assayed following the method of Cormier et al. [50] where the enzyme catalyzes the oxidation of NADH + H^+^ to NAD^+^, with the ultimate reduction of ubiquinone to ubiquinol. The rate of UBIQ-RD activity was measured as a rotenone-sensitive rate of NADH oxidation at 37 °C and 340 nm.

### 4.7. Markers of Cellular Antioxidant Homeostasis 

Total antioxidant status (TAS) was measured using a kit from RANDOX Laboratories (Crumlin, Co., Antrim, UK). ABTS^®^ (2,2′-Azino-di-[3-ethylbenzthiazoline sulphonate]) was incubated with a peroxidase (metmyoglobin) and H_2_O_2_ to produce the free radical cation ABTS^®+^. This has a relatively stable blue-green color, which is measured at 600 nm. Antioxidants in the sample cause suppression of this color production in proportion to their concentration [51].

γ-Glutamylcysteine synthetase (γ-GCS) activity was determined from the rate of formation of ADP (assumed to be equal to the rate of oxidation of NADH) calculated from the change in absorbance at 340 nm [52]. The above-mentioned colorimetric assays were adapted for use on the KONLAB clinical chemistry analyzer (Thermo Clinical LabSystems, Espoo, Finland).

### 4.8. Markers of Oxidative Damage 

Protein carbonyls (PC) were assayed using commercial kits from Cayman Chemical Company (Ann Arbor, MI). This assay kit utilizes the 2,4-dinitrophenylhydrazine (DNPH) reaction to measure the protein carbonyl content in a 96-well format. The amount of protein-hydrozone produced was quantified at an absorbance of 370 nm.

The activity of total aconitase (cytosolic and mitochondrial), based on the formation of NADPH from NADP^+^, was assayed using commercial kits (OXIS International Inc., Portland, OR, USA). Citrate was converted to isocitrate (catalyzed by aconitase), which then underwent oxidative decarboxylation (catalyzed by isocitrate dehydrogenase) and became α-ketoglutarate. Concomitantly during this last reaction, NADP+ was reduced to NADPH. NADPH formation was measured at 340 nm absorbance, and was proportional to aconitase activity.

### 4.9. Statistical Analysis

For each outcome variable, two-way ANOVAs were run separately for each study (acute and subacute) by brain region combination. First, a factorial ANOVA was run, looking at effects of ozone, diet, and any interaction of the two. If there was an interaction (*p* < 0.05), all pairwise t-tests among the O_3_ by diet groups were examined. If there was no interaction (*p* ≥ 0.05), the interaction term was removed and the main effects model run. If there was a significant O_3_ and/or diet effect (*p* < 0.05) in the main effects model, pairwise tests of differences between the ozone and control animals, and/or among the diet groups, were tested. The ANOVA assumptions of normality and homogeneity of variance were examined using scatterplots of data and model residuals via the Shapiro–Wilk test (normality) and Levene’s test (homogeneity of variance). A log transformation was used when it improved normality and/or homogeneity of variance, as was the case with Aconitase, γ-GCS, and TAS. As this was an exploratory study, many analyses and tests were done with no adjustments for multiple comparisons. Therefore, individual significant tests should be interpreted with caution, and emphasis placed on patterns across the data rather than individual tests.

## Figures and Tables

**Figure 1 ijms-20-00011-f001:**
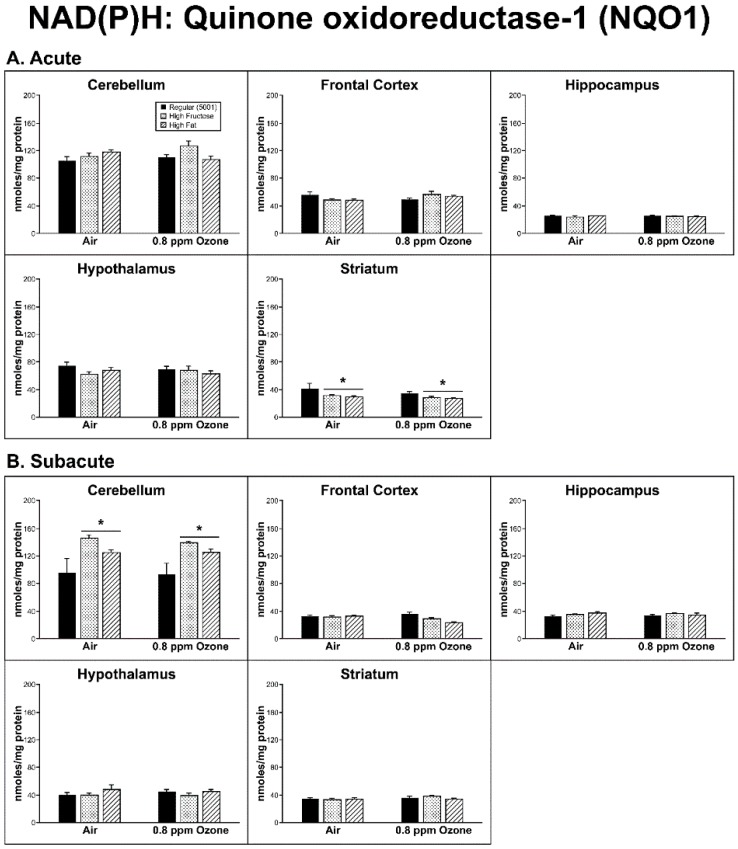
Effects of (**A**) acute and (**B**) subacute ozone exposure on NAD(P)H: Quinone oxidoreductase (NQO1) activity in cerebellum, frontal cortex, hippocampus, hypothalamus, and striatum of Brown Norway male rats maintained on either a regular, high-fructose, or high-fat diet. Each value is a mean ± SE of five rats except for: Acute—Hypothalamus—0.8 ppm—High Fat (*n* = 4). * *p* < 0.05, within diet compared to Regular (5001).

**Figure 2 ijms-20-00011-f002:**
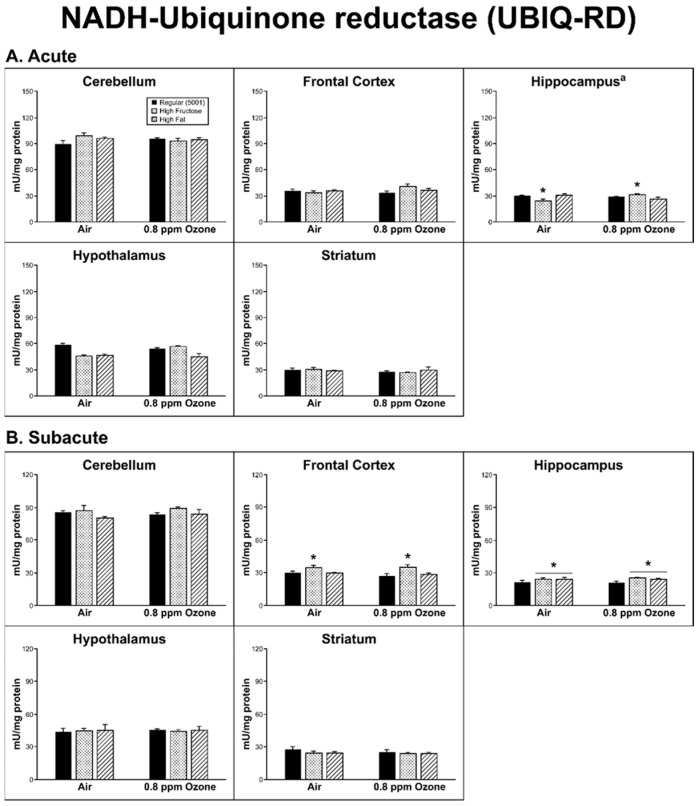
Effects of (**A**) acute and (**B**) subacute ozone exposure on NADH-ubiquinone reductase (UBIQ-RD) activity in the cerebellum, frontal cortex, hippocampus, hypothalamus, and striatum of Brown Norway male rats maintained on either a regular, high-fructose, or high-fat diet. Each value is a mean ± SE of five rats except for: Hypothalamus—0.8 ppm—High Fat (*n* = 4). * *p* < 0.05, within diet treatment compared to Regular (5001); ^a^ Significant interaction between dietary condition and ozone treatment, *p* < 0.01.

**Figure 3 ijms-20-00011-f003:**
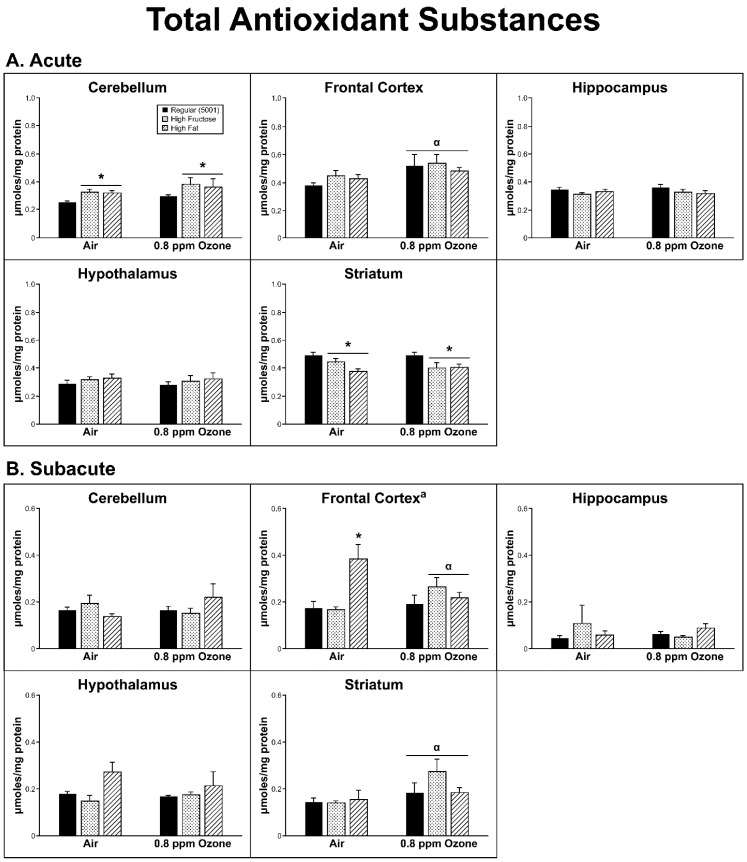
Effects of (**A**) acute and (**B**) subacute ozone exposure on total antioxidant substances (TAS) in cerebellum, frontal cortex, hippocampus, hypothalamus, and striatum of Brown Norway male rats maintained on either a regular, high-fructose, or high-fat diet. Each value is a mean ± SE of five rats except for: Acute—Hypothalamus—0.8 ppm—High Fat (*n* = 4); Subacute—Hippocampus—Air/0.8 ppm—Regular (*n* = 4). α: *p* < 0.05, within diet compared to Air; * *p* < 0.05, within ozone treatment compared to Regular (5001); ^a^ Significant interaction between dietary condition and ozone treatment, *p* < 0.01.

**Figure 4 ijms-20-00011-f004:**
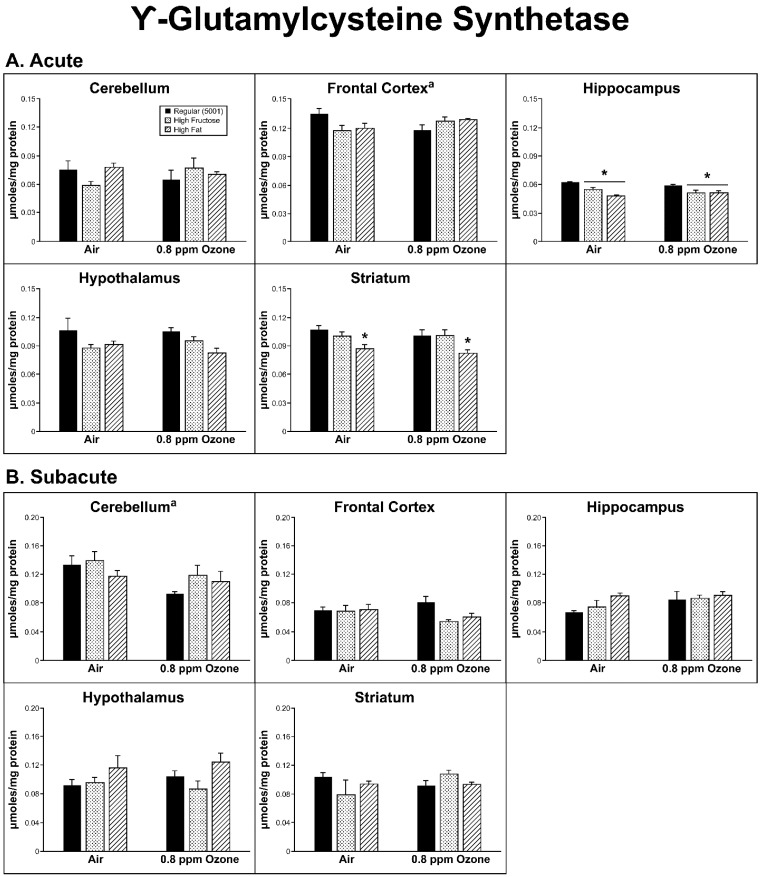
Effects of (**A**) acute and (**B**) subacute ozone exposure on γ-Glutamylcysteine synthetase (γ-GCS) activity in the cerebellum, frontal cortex, hippocampus, hypothalamus, and striatum of Brown Norway male rats maintained on either a regular, high-fructose, or high-fat diet. Each value is a mean ± SE of five rats except for: Hypothalamus—0.8 ppm—High Fat (*n* = 4). * *p* < 0.05, within diet treatment compared to Regular (5001). ^a^ Significant interaction between dietary condition and ozone treatment, *p* < 0.05.

**Figure 5 ijms-20-00011-f005:**
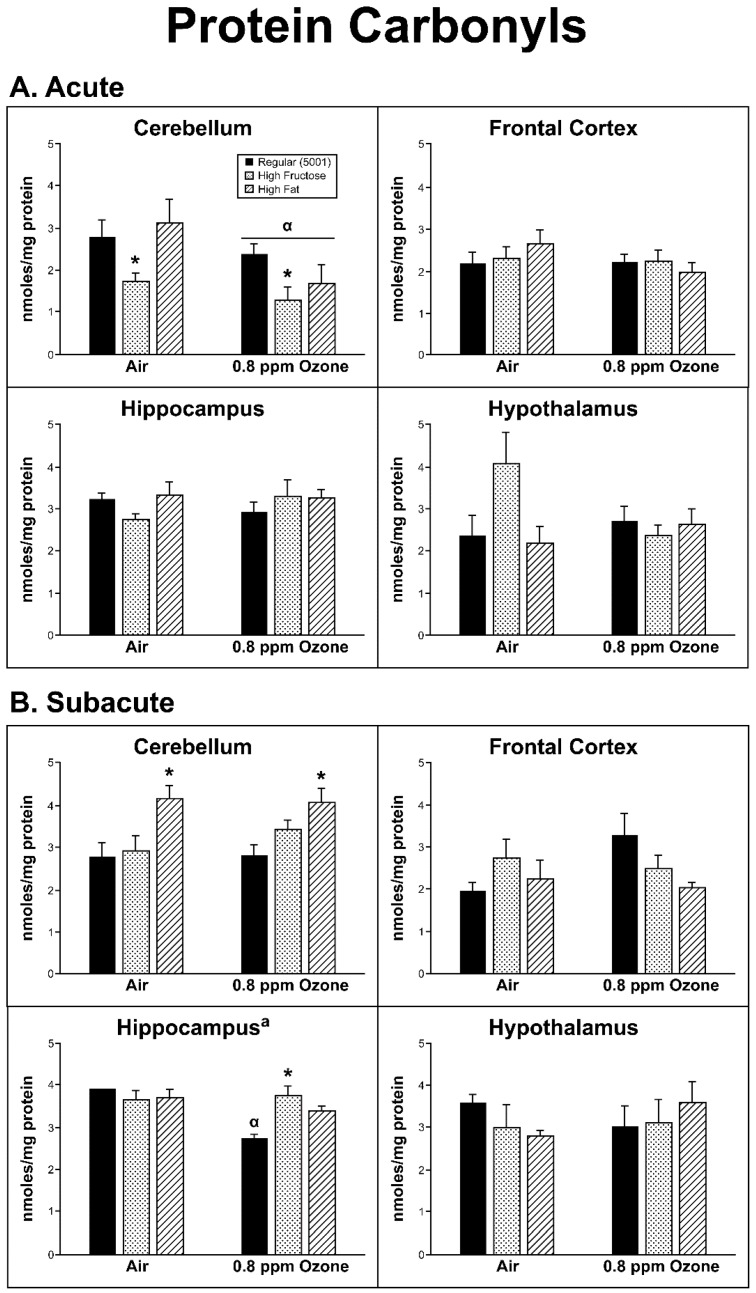
Effects of (**A**) acute and (**B**) subacute ozone exposure on protein carbonyl content in cerebellum, frontal cortex, hippocampus, and hypothalamus of Brown Norway male rats maintained on either a regular, high-fructose, or high-fat diet. Each value is a mean ± SE of five rats except for: Acute—Hypothalamus—0.8 ppm—High-Fructose (*n* = 4); Subacute—Cerebellum—Air—Regular (*n* = 8); Subacute—Hippocampus—Air/0.8 ppm—Regular (*n* = 4); Subacute—Hypothalamus—Air/0.8 ppm—Regular (*n* = 7/8). α: *p* < 0.05, within diet compared to Air; * *p* < 0.05, within diet treatment compared to Regular (5001). ^a^ Significant interaction between dietary condition and ozone treatment, *p* < 0.01.

**Figure 6 ijms-20-00011-f006:**
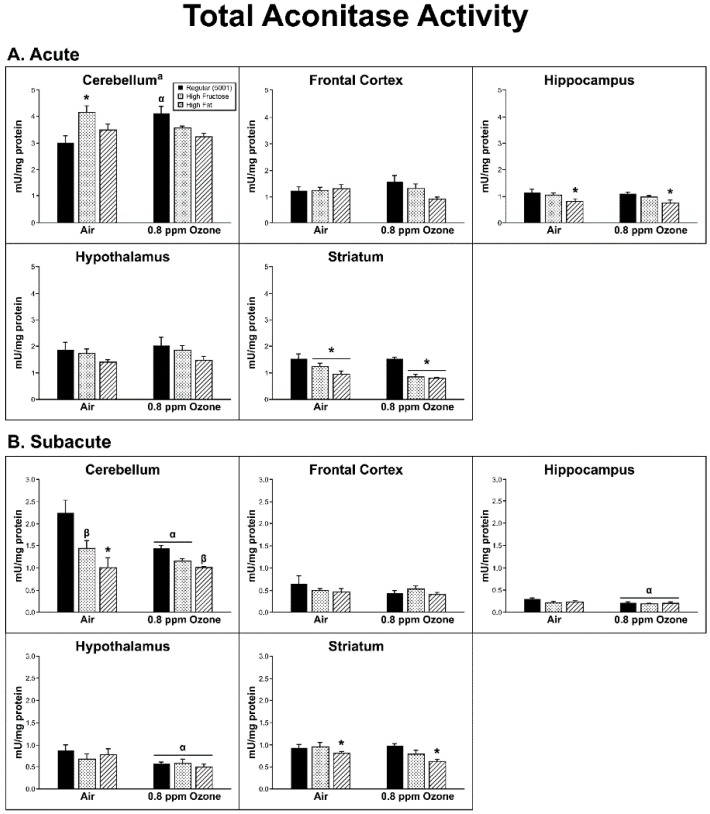
Effects of (**A**) acute and (**B**) subacute ozone exposure on total aconitase activity in cerebellum, frontal cortex, hippocampus, hypothalamus, and striatum of Brown Norway male rats maintained on either a regular, high-fructose, or high-fat diet. Each value is a mean ± SE of five rats. α: *p* < 0.05, within diet compared to Air; *: *p* < 0.05, within ozone treatment compared to Regular (5001); β*: *p* < 0.01, within ozone treatment compared to Regular (5001). ^a^ Significant interaction between dietary condition and ozone treatment, *p* < 0.01.

**Figure 7 ijms-20-00011-f007:**
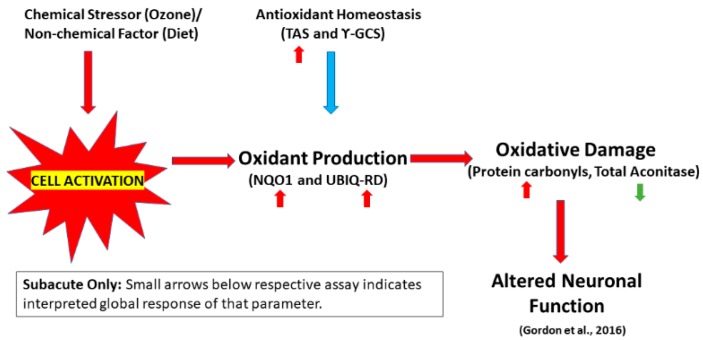
Schematic of interpreted overall effects following subacute exposures to O_3_ and diets. Arrows below respective assay indicate an increase (red arrow) or decrease (green arrow) in that particular parameter. Big red arrow indicates negative effect on the cell while Big blue arrow is indicative of a positive effect and when antioxidant homeostasis is unbalanced with oxidant production, oxidative damage occurs.

**Figure 8 ijms-20-00011-f008:**
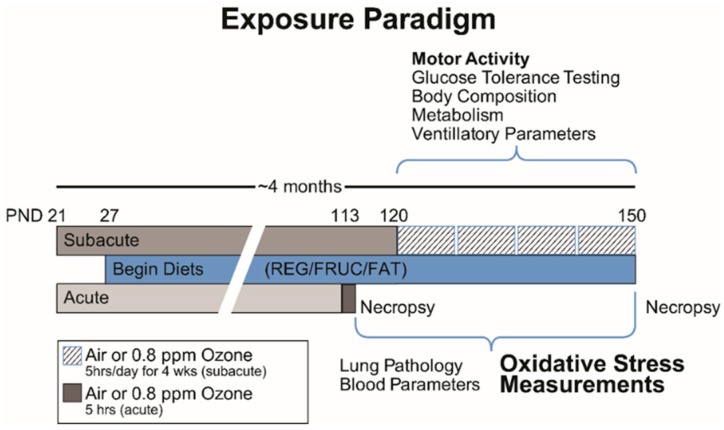
Exposure Paradigm. Animals were received on postnatal day (PND) 21 and either control, high-fat, or high-fructose diets were started on PND 27. Subjects remained on diet throughout the entirety of the study. Acute group was exposed once to O_3_ on PND 113 and terminated 18 h following. Subacute group received four O_3_ exposures over 4 weeks and terminated 18 h after final exposure. During this time, physiological parameters (motor activity, glucose tolerance, body composition, metabolism, and ventilatory function) were examined. Following termination, both groups were processed for pathology and brain regions were collected. Timeline is not to scale.

**Table 1 ijms-20-00011-t001:** Composition of diets (% by weight).

Component	Regular (5001) ^a^	High Fructose ^b^	High Fat ^c^
Fat	13.5	5.2	34.3
Carbohydrates	58.0	60.4	27.3
Protein	28.5	18.3	23.5
Metabolizable Energy (kcal/g)	3.02	3.6	5.1

^a^ Ralston Purina; ^b^ Harlan TD.89247; ^c^ Harlan TD.06414.

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
