# Peer review of "Interaction of Diet and Ozone Exposure on Oxidative Stress Parameters within Specific Brain Regions of Male Brown Norway Rats"

_ijms, 2018, doi:10.3390/ijms20010011_

Round 1
Reviewer 1 Report
In this study, the authors examined the interaction of diet and ozone exposure on oxidative stress parameters within Specific Brain Regions of male Brown Norway rats. It is important from the clinical perspective to know the mechanism of action. However, there are problems the way of presented data. It is quite hard to understand that what authors wanted to describe. I recommend to write the schema of discussion part including authors results.
Author Response
REVIEWER #1:
Comment 1: In this study, the authors examined the interaction of diet and ozone exposure on oxidative parameters within specific brain regions of male Brown Norway rats. It is important from the clinical perspective to know the mechanism of action. However, there are problems the way of presented data. It is quite hard to understand that what authors want to describe. I recommend to write the schema of discussion part including authors results.
AUTHOR’S RESPONSE: We thank the Reviewer for the thorough review and for considering the importance of this work. We do agree that the experimental design is very complex and the results are difficult to interpret as we have so many variables. As suggested, we have prepared a schematic displaying the results (See new Figure 7).
Reviewer 2 Report
This study explores the interactive effects of O3 and diet (high fructose or high fat) on oxidative stress in different rat brain regions. In acute exposure, there was a decrease in markers of ROS production in some brain regions by diet and not by O3. In general, these results indicate that diet/O3 did not have a global effect on brain oxidative stress parameters, but showed some brain region‐ and oxidative stress parameter‐specific effects by diets. In this study, five brain regions were assayed for 5 markers of oxidative stress after O3 exposure and/or on the diets which correlate with metabolic disorders to investigate the interaction of diets and O3 exposure on oxidative stress. However, it is difficult to make a conclusion on this issue according the experimental design.
Major points:
1. The interaction of diets and O3 exposure on oxidative stress is compared among brain regions in five figures. However, the effects among assays is not comparable. Hence, a table is required to summarize and compare the assays in all brain regions, and which will be helpful to interpret the results more deeply.
2. Some assays on neuroinflammation may be included since neuroinflammation is highly correlated with oxidative stress.
3. Striatum was missed in protein carbonyls assay. For comparison with the other assays, the assay of this region should be included.
4. In Figure 6, what is the “b” mean on statistical analysis.
Minor points:
1. In Abstract section: Most of abbreviations such as OS, FRUC, FAT, CER, FC, STR, PC, HIP, are not necessary, should be full spelled.
2. In text: Most of abbreviations such as SF, HFCS, OS, HYP, CER, STR, FC, are not necessary, should be full spelled.
3. In text: full spelling of the most of abbreviations such as CNS, NQO1, UBIQ-RD, TAS, should be displayed in “introduction section” or “Result section” but not in “Discussion section” or “Materials and method section”.
4. The name of groups should be consistent. For example, it is inconsistent for FAT/control, FRUC/control, Regular diet/O3, FRUC/O3,described in page 7 and 2 as compared with FAT/0, Reg/0.8, Reg/0 described in page 9,
5. Page 2, Line 9, “sympathetic nervous system” should be “SNS”.
6. Page 2, Line 13, “Ozone (O3)” should be “O3”.
7. Page 2, Line 30, “hypothalamic-pituitary adrenal (HPA)” should be “HPA”.
8. Page 4, Line 2, “NADH Ubiquinone reductase” should be “NADH Ubiquinone reductase (UBIQ-RD)”.
9. Page 6, Line 9, “(GCS)” should be “(γ-GCS)”.
10. Page 13, Line 3, “PND” should be “postnatal day (PND)”.
11. Page 13, Line 3, “PND 120” should be “PND 113”.
Author Response
REVIEWER #2:
COMMENT 1: This study explores the interactive effects of O3 and diet (high fructose or high fat) on oxidative stress in different rat brain regions. In acute exposure, there was a decrease in markers of ROS production in some brain regions by diet and not by O3. In general, these results indicate that diet/O3 did not have a global effect on brain oxidative stress parameters, but showed some brain region‐ and oxidative stress parameter‐specific effects by diets. In this study, five brain regions were assayed for 5 markers of oxidative stress after O3 exposure and/or on the diets which correlate with metabolic disorders to investigate the interaction of diets and O3 exposure on oxidative stress. However, it is difficult to make a conclusion on this issue according the experimental design.
AUTHOR’S RESPONSE: We understand that the experimental design is very complex with many variables. However, this is an important aspect where we are trying to understand the influence of diet on environmental chemical exposures. As you are aware, childhood obesity is on the rise globally and these studies are important to understand in terms of chemical effects on communities.
COMMENT: Major points:
The interaction of diets and O3 exposure on oxidative stress is compared among brain regions in five figures. However, the effects among assays is not comparable. Hence, a table is required to summarize and compare the assays in all brain regions, and which will be helpful to interpret the results more deeply.
AUTHOR’S RESPONSE: Comparing assays is not the focus of this study. We are using these assays to understand oxidant production, antioxidant homeostasis, and protein damage following exposure to ozone and diet. For sake of clarity, we have presented this information in the schematic (new Figure 7).
Some assays on neuroinflammation may be included since neuroinflammation is highly correlated with oxidative stress.
AUTHOR’S RESPONSE: We agree with the reviewer that neuroinflammation is highly related to oxidative stress. We will consider this in our future experiments. Thanks for the suggestion.
Striatum was missed in protein carbonyls assay. For comparison with the other assays, the assay of this region should be included.
AUTHOR’S RESPONSE: We did not have enough striatum to do the protein carbonyls as we have used all the tissue for assays related to oxyradical production and antioxidants. As reviewers are aware, striatal tissue is very small in less than 25 mg as compared to other brain regions.
4. In Figure 6, what is the “b” mean on statistical analysis.
AUTHOR’S RESPONSE: I think that this reviewer means “β(beta)” instead of “b”. It is explained in the figure legends as β means; p < 0.01, within ozone treatment compared to Regular (5001) diet.
COMMENT: Minor points:
In Abstract section: Most of abbreviations such as OS, FRUC, FAT, CER, FC, STR, PC, HIP, are not necessary, should be full spelled.
AUTHOR’S RESPONSE: Thanks for bringing this to our attention. Unfortunately, we are limited to the number of characters as per journal format and hence we used abbreviations in the abstract.
In text: Most of abbreviations such as SF, HFCS, OS, HYP, CER, STR, FC, are not necessary, should be full spelled.
AUTHOR’S RESPONSE: Thanks for bringing this to our attention. We are following the journal format and these abbreviations were used many times and they were spelled when they are used for the first time as per instructions in manuscript preparation.
In text: full spelling of the most of abbreviations such as CNS, NQO1, UBIQ-RD, TAS, should be displayed in “introduction section” or “Result section” but not in “Discussion section” or “Materials and method section”.
AUTHOR’S RESPONSE: Thanks for bringing this to our attention. We are following the journal format and these abbreviations were used many times and they were spelled when they are used for the first time as per instructions in manuscript preparation.
The name of groups should be consistent. For example, it is inconsistent for FAT/control, FRUC/control, Regular diet/O3, FRUC/O3,described in page 7 and 2 as compared with FAT/0, Reg/0.8, Reg/0 described in page 9,
AUTHOR’S RESPONSE: Thanks for pointing this. We have now corrected.
Page 2, Line 9, “sympathetic nervous system” should be “SNS”.
AUTHOR’S RESPONSE: Thanks for pointing this. We have now corrected
Page 2, Line 13, “Ozone (O3)” should be “O3”.
AUTHOR’S RESPONSE: Thanks for pointing this. We have now corrected
Page 2, Line 30, “hypothalamic-pituitary adrenal (HPA)” should be “HPA”.
AUTHOR’S RESPONSE: Thanks for pointing this. We have now corrected.
Page 4, Line 2, “NADH Ubiquinone reductase” should be “NADH Ubiquinone reductase (UBIQ-RD)”.
AUTHOR’S RESPONSE: Thanks for pointing this. We have now corrected.
Page 6, Line 9, “(GCS)” should be “(γ-GCS)”
AUTHOR’S RESPONSE: Thanks for pointing this. We have now corrected.
Page 13, Line 3, “PND” should be “postnatal day (PND)”.
AUTHOR’S RESPONSE: Thanks for pointing this. We have now corrected.
Page 13, Line 3, “PND 120” should be “PND 113”.
AUTHOR’S RESPONSE: Thanks for pointing this. We have now corrected.
Reviewer 3 Report
In this manuscript entitled “Interaction of Diet and Ozone Exposure on Oxidative Stress Parameters within Specific Brain Regions of Male Brown Norway Rats,” the authors investigated the effects of ozone and diet [high fructose (FRUC) or high fat (FAT)] on oxidative stress-related markers. This manuscript includes interesting findings; however, the logic of this research is unclear and the experimental design is not well described. My major comments are as follows:
Major comments:
1. It is unclear why the authors investigated the effects of ozone and diet. The authors should describe the meaning of this experiments in the introduction, clearly. In addition, the authors should express this essence in the abstract, briefly.
2. In the results section, the authors should show the experimental design of this study at first. Please move Fig.7 to Fig.1. In addition, the basic information such as body weight and blood glucose and/or lipids of rats are lacked.
3. The meanings of ozone levels are also unclear. Why did the authors select this condition?
Author Response
REVIEWER #3:
In this manuscript entitled “Interaction of Diet and Ozone Exposure on Oxidative Stress Parameters within Specific Brain Regions of Male Brown Norway Rats,” the authors investigated the effects of ozone and diet [high fructose (FRUC) or high fat (FAT)] on oxidative stress-related markers. This manuscript includes interesting findings; however, the logic of this research is unclear and the experimental design is not well described. My major comments are as follows:
AUTHOR’S RESPONSE: The purpose of this study was clearly mentioned in the last paragraph of Introduction. The logic behind this research is that obesity due to certain diets is increasing throughout the world and they can be exposed to air pollutants. Here we are trying to understand the diet effects on air pollution toxicity.
Major comments:
It is unclear why the authors investigated the effects of ozone and diet. The authors should describe the meaning of this experiments in the introduction, clearly. In addition, the authors should express this essence in the abstract, briefly.
AUTHOR’S RESPONSE: The purpose of this study was clearly mentioned in the last paragraph of Introduction. The logic behind this research is that obesity due to certain diets is increasing throughout the world and they can be exposed to air pollutants. Here we are trying to understand the diet effects on air pollution toxicity where this information is needed to determine the risk of those individuals to air pollutants.
In the results section, the authors should show the experimental design of this study at first. Please move Fig.7 to Fig.1. In addition, the basic information such as body weight and blood glucose and/or lipids of rats are lacked.
AUTHOR’S RESPONSE: We agree with this reviewer that experimental design should be figure 1. However, the journal format dictates that methods section is at the end, hence our figure 1 became figure 7. With the new figure, Figure 7 now became Figure 8. The information on body weight, blood glucose and lips were published previously from the same cohort of animals [see Gordon et al…Inhalation Toxicol. 28, 203-215 (2016)].
The meanings of ozone levels are also unclear. Why did the authors select this condition?
AUTHOR’S RESPONSE: Ozone is one of the six criterial air pollutants. National Air Quality standards are set at 0.07 ppm for 8 hour daily exposure, however we used 0.8 ppm for our study because of less exposure time and we have used lower doses also.
Round 2
Reviewer 1 Report
The authors have responded appropriately to our original concerns.
Reviewer 2 Report
It is acceptable after revision.
Reviewer 3 Report
This reviewer could not fully satisfy the authors' response to my comments. Therefore, it is not possible to recommend the publication of this paper. The composition of this paper is unkind to the reader, and the meaning and purpose of each experiment were difficult to understand.